# Transport and inhibition of the sphingosine-1-phosphate exporter SPNS2

Huanyu Z. Li [1], Ashley C. W. Pike [1,9], Yung-Ning Chang[2,9], Dheeraj Prakaash[3,9], Zuzana Gelova[4,9], Josefina Stanka [5,9], Christophe Moreau[1], Hannah C. Scott[1,6], Frank Wunder [5], Gernot Wolf [4], Andreea Scacioc[1], Gavin McKinley[1], Helena Batoulis[5], Shubhashish Mukhopadhyay[1], Andrea Garofoli[4], Adán Pinto-Fernández [1,6], Benedikt M. Kessler [1,6], Nicola A. Burgess-Brown[1], Saša Štefanić [7], Tabea Wiedmer[4], Katharina L. Dürr [1] ✉, Vera Puetter[2] ✉, Alexander Ehrmann[5] ✉, Syma Khalid [3] ✉, Alvaro Ingles-Prieto[4] ✉, Giulio Superti-Furga [4,8] ✉ & David B. Sauer [1] ✉

Sphingosine-1-phosphate (S1P) is a signaling lysolipid critical to heart development, immunity, and hearing. Accordingly, mutations in the S1P transporter SPNS2 are associated with reduced white cell count and hearing defects. SPNS2 also exports the S1P-mimicking FTY720-P (Fingolimod) and thereby is central to the pharmacokinetics of this drug when treating multiple sclerosis. Here, we use a combination of cryo-electron microscopy, immunofluorescence, in vitro binding and in vivo S1P export assays, and molecular dynamics simulations to probe SPNS2's substrate binding and transport. These results reveal the transporter's binding mode to its native substrate S1P, the therapeutic FTY720-P, and the reported SPNS2-targeting inhibitor 33p. Further capturing an inward-facing apo state, our structures illuminate the protein's mechanism for exchange between inward-facing and outward-facing conformations. Finally, using these structural, localization, and S1P transport results, we identify how pathogenic mutations ablate the protein's export activity and thereby lead to hearing loss.

Sphingosine-1-phosphate (S1P) is a bioactive lipid central to cell growth, embryonic development, and the physiology and pathophysiology in multiple tissues. S1P is essential to vascular and cardiac development[1,2], and maintenance of the blood-brain barrier[3]. The S1P concentration gradient in peripheral lymphoid organs is critical for the migration of lymphocytes[4]. Accordingly, the S1P signaling pathway is targeted by several clinically approved or evaluated therapeutics for auto-immune diseases including multiple sclerosis and ulcerative colitis[5,6]. Finally, while its role is complex, the lipid is central to cancer by regulating vascularization, inflammation, and cell growth[7].

Sphingosine-1-phosphate is generated from sphingosine in the cytoplasm by the sphingosine kinases[8]. The signaling lipid is subsequently exported to the extracellular space by several plasma membrane export proteins, including Spinster Homolog 2 (SPNS2),

[1]Centre for Medicines Discovery, Nuffield Department of Medicine, University of Oxford, Oxford, UK. [2]Nuvisan ICB GmbH, Berlin, Germany. [3]Department of Biochemistry, University of Oxford, Oxford, UK. [4]CeMM Research Center for Molecular Medicine of the Austrian Academy of Sciences, Vienna, Austria. [5]Bayer AG, Leverkusen, Germany. [6]Chinese Academy for Medical Sciences Oxford Institute, Nuffield Department of Medicine, University of Oxford, Oxford, UK. [7]Nanobody Service Facility, University of Zurich, AgroVet-Strickhof, Lindau, Switzerland. [8]Center for Physiology and Pharmacology, Medical University of Vienna, Vienna, Austria. [9]These authors contributed equally: Ashley C. W. Pike, Yung-Ning Chang, Dheeraj Prakaash, Zuzana Gelova, Josefina Stanka. ✉e-mail: katharina.duerr@omass.com; Vera.Puetter@nuvisan.com; alexander.ehrmann@bayer.com; syma.khalid@bioch.ox.ac.uk; ingles.prieto@gmail.com; GSuperti@cemm.oeaw.ac.at; david.sauer@cmd.ox.ac.uk

MFSD2B, ABCA1, ABCA7, ABCC1, and ABCG2[9]. Of these, SPNS2 is the primary exporter of S1P into lymph[10]. Accordingly, knock-out of SPNS2 in mice prevents immune cell egress into lymph and blood[10], and a single nucleotide variant is associated with white cell count in humans[11]. Additionally, loss of SPNS2 function leads to hearing defects in humans and mice due to disorganization of the stria vascularis and loss of the endocochlear potential[12,13].

Notably, SPNS2 also plays a central role in the therapeutic targeting of sphingosine-1-phosphate signaling. The S1P receptor (S1PR) antagonist fingolimod is administered as the prodrug FTY720 and phosphorylated intracellularly into the bioactive FTY720 phosphate (FTY720-P)[14]. Exported by SPNS2, FTY720-P subsequently induces internalization and downregulation of S1P receptors[15], ultimately leading to lymphopenia[16]. However, the S1PR antagonists have cardiovascular side effects due to receptor expression in cardiac cells[17]. Therefore, alternative methods of targeting S1P signaling are of significant interest[18], and SPNS2 itself has been the target for small-molecule inhibitor development[19,20].

Historically, understanding of SPNS2's binding to substrates and inhibitors has been limited by a lack of structural data for the transporter, challenging structure-based drug design. Several distantly related transporters from the larger major facilitator superfamily serve as prototypes for MFS lipid transport[21,22], including the lysophospholipid importer MFSD2A[23–26] and the orphan transporter HnSPNS from *Hyphomonas neptunium*[27]. Recently, apo structures of human SPNS2 in an outward-facing conformation ($C_o$-apo), and inward-facing structures in complex with S1P ($C_i$-S1P) and the reported inhibitor 16d ($C_i$-16d), have shed light on the protein's transport and inhibition[28]. However, the mechanism of substrate entry to the binding site, and the outward-to-inward conformational change, remain ambiguous without an inward-facing apo state. Further, positioning of the S1P head group in the substrate-bound SPNS2 structure would preclude a classic MFS rocker-switch mechanism[29], necessitating further investigation of substrate binding and transport. Finally, the transporter's interactions with FTY720-P and the reported high-potency SPNS2 inhibitor SLB1122168 (33p) remain undescribed.

To address these gaps in our knowledge of SPNS2 transport and enable therapeutic targeting of the transporter, we determine cryo-electron microscopy (cryo-EM) of inward-facing SPNS2 in an apo state and with a detergent occupying the binding site. In conjunction with these structural results, we use molecular dynamics simulations and assessed biochemical thermostability to characterize the transporter's interactions with S1P, FTY720-P, and a SPNS2-targeting inhibitor. Finally, we report a S1PR3-coupled S1P transport assay amenable for high-throughput screening of SPNS2 activity. Together, these provide valuable insights into SPNS2's transport activity and enable its therapeutic targeting.

## Results

### Anti-SPNS2 nanobody generation and subcellular localization
To enable nanobody generation, biophysical, and structural studies of SPNS2, we first over-expressed the protein in HEK293 cells using the BacMam system and purified the protein to homogeneity as a monomeric species in n-dodecyl-β-D-maltopyranoside (DDM) or lauryl maltose neopentyl glycol (LMNG) (Supplementary Fig. 1a–c). The solubilized SPNS2 was thermostabilized by FTY720-P in a dose-dependent manner up to 2 °C (Fig. 1a, Supplementary Fig. 1d). However, FTY720-P did not stabilize another major facilitator transporter, the voltage-gated purine nucleotide uniporter (VNUP) (Supplementary Fig. 1e). This indicates that solubilized and purified SPNS2 is properly folded and capable of binding substrate through specific interactions with the transporter.

Immunizing alpacas with SPNS2, we identified nanobodies D12 (NbD12) and F09 (NbF09) which bind SPNS2 with affinities of 7.75 nM and 368 nM, respectively (Supplementary Fig. 1f, Fig. 1b). To further validate the nanobodies in cells, and test their applicability for in vivo studies, we took advantage of a HEK293 cell line with inducible overexpression of SPNS2 fused to an HA tag. With negative and positive controls of cells without induction or induced cells, we found that SPNS2 localizes to the plasma membrane and both NbD12 and NbF09 co-localize with the transporter (Fig. 1c, Supplementary Fig. 1g). Confirming their ability to bind solubilized SPNS2, co-immunoprecipitation with the nanobodies from cells pulled down the transporter from expressing cells (Supplementary Fig. 1h). These results confirm that NbD12 and NbF09 are highly specific for SPNS2 and bind the transporter in detergent and its native membrane.

### Inward-facing structure of SPNS2
We next set out to determine an experimental structure of full-length SPNS2 to examine its binding and transport of S1P. However, the small size of SPNS2 presented a challenge for single-particle cryo-EM. As NbD12 bound SPNS2 with higher affinity than NbF09, we prepared and purified the SPNS2-NbD12 complex (Supplementary Fig. 3a, b) and determined its structure by cryo-EM (Fig. 1d, Supplementary Fig. 3c, Supplementary Table 1). This 3.7 Å resolution map of the SPNS2-NbD12 complex in DDM (SPNS2-DDM) was sufficient to build and refine the entire transporter except for residues 1–99 from the highly mobile N-terminus, and residues 287–296 and 352–358 in extramembrane loops L6-7 and L7-8, respectively (Fig. 1e).

As expected for an MFS transporter, SPNS2 is composed of 12 transmembrane helices organized into pseudo symmetric N- and C-domains of TM1-6 and TM7-12 (Supplementary Fig. 4a, b). Notably, TM11 is broken by a well-resolved pi helix (Supplementary Fig. 4c), with a similar feature seen only in structures of the orphan HnSPNS and SLCO6C1 proteins[27,30]. An intracellular helix (ICH1) is found N-terminal of TM7, and a second intracellular helix (ICH2) is located immediately after TM12. Nanobody D12 engages with the cytoplasmic face of the C-domain of SPNS2, with most contacts involving ICH2 and additional interactions with loops L8-9 and L10-11. Notably, the single-particle cryo-EM analysis did not show evidence of structural heterogeneity or alternative conformations of the transporter (Supplementary Fig. 3c), suggesting NbD12 favors the inward-facing conformation.

The transporter adopts a classic inward-facing MFS conformation, with the central binding site open to the cytoplasm (Fig. 1e), similar to the substrate and inhibitor-bound SPNS2 structures (RMSD = 0.51–0.58 Å). The extracellular gate is sealed by opposing pairs of hydrophobic residues on TM1 and TM7, with Tyr246 making a hydrogen bond with the carbonyl of Gly333 (Fig. 1f). This tyrosine-carbonyl hydrogen bond appears conserved among Spinster family transporters, with the equivalent of Tyr246 and Gly333 conserved as tyrosine and alanine or glycine respectively (Supplementary Fig. 2). Beyond this hydrophobic layer, a network of polar residues from TM1b, TM2, TM7, and TM11b form hydrogen bonds and salt bridges which further stabilize the closed extracellular gate.

### Structure of inward-facing DDM-bound SPNS2
Within the SPNS2 Coulombic potential map, there is an unexpected density within the transporter that is unexplained by the protein model (Fig. 2a, Supplementary Fig. 4d). This molecule appears amphipathic, extending from the central cavity into a hydrophobic pocket within the C-domain of SPNS2. While S1P co-purified with SPNS2 after digitonin extraction[28], native mass spectrometry of the DDM-purified SPNS2 did not identify co-purified S1P[31]. As DDM and sphingosine-1-phosphate are amphipathic molecules with a single acyl chain, we hypothesize the detergent can occupy the native substrate's binding site. Supporting this hypothesis, the unknown density fits the acyl chain and first glucose of n-dodecyl-β-D-maltopyranoside. Therefore, we modeled this density as a dodecyl glucoside molecule (Fig. 2b), though we cannot exclude a mixture of DDM and S1P.

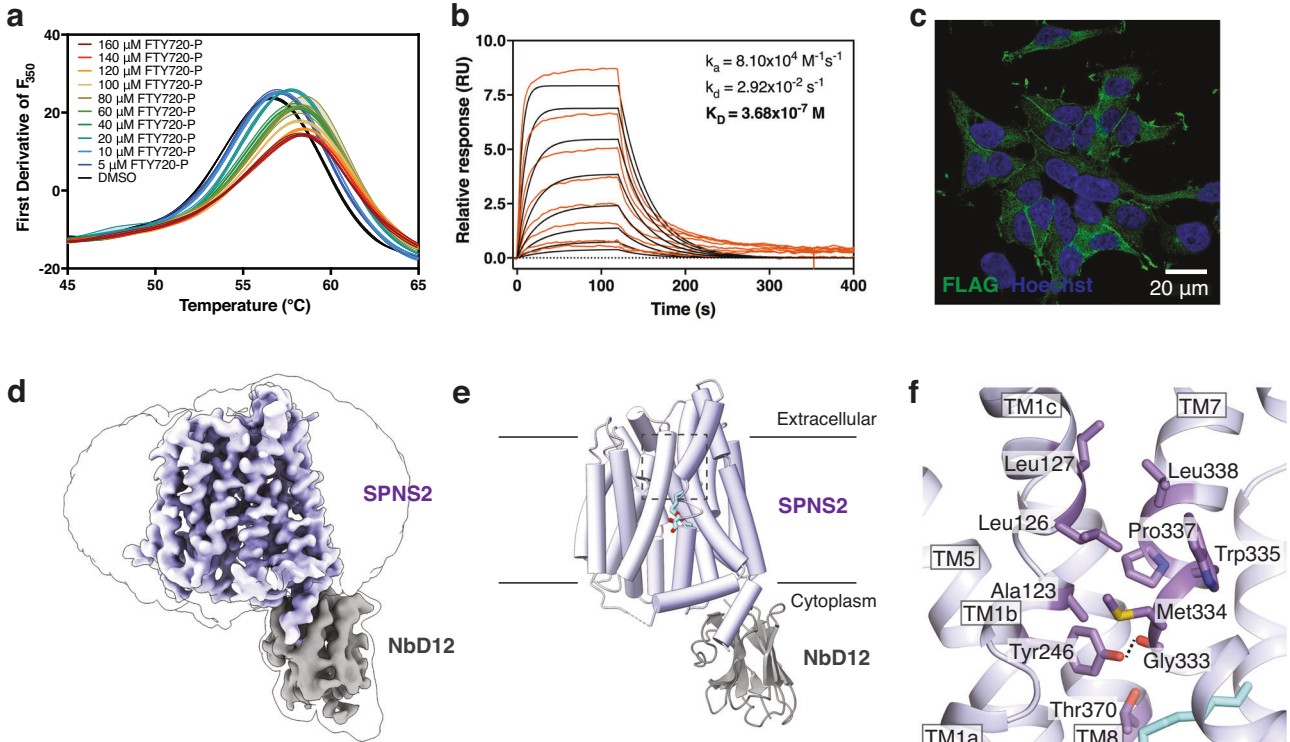

**Fig. 1 | Structure of detergent-solubilized SPNS2. a** Measurement of LMNG-purified SPNS2 by FTY720-P using nano-Differential Scanning Fluorescence (N = 3). Data are presented as mean ± SEM. **b** Binding affinity and kinetics of NbF09 as determined by Surface Plasmon Resonance. Injections at various concentrations and best fits are shown as orange and black lines, respectively. **c** Anti-FLAG immunofluorescence of SPNS2 overexpressing cells stained with FLAG-tagged NbF09 and Hoechst dye. Representative image with clear signal of the relevant immunofluorescence channel and appropriate Z plane after collecting 3–5 images from different fields of biological duplicates. **d** Cryo-EM map of the SPNS2-NbD12 complex determined in DDM. Transporter and nanobody are colored in purple and gray, respectively. The detergent micelle and extent of the entire molecule are shown by a thin black outline (blurred version of the same cryo-EM map). **e** Structure of the SPNS2-NbD12 complex determined in DDM viewed from the membrane plane. The region of the extracellular gate is indicated with a dotted box and shown in greater detail in the adjacent panel. **f** Residues sealing the extracellular gate of SPNS2. Hydrogen bonds are shown as dotted lines.

Inspecting this inward-facing DDM-bound ($C_i$-DDM) structure of SPNS2, we noticed the n-dodecyl-β-D-maltopyranoside primarily contacts the C-domain, with its acyl tail partially inserted into a pocket between TM7, TM8, and TM10 (Fig. 2b, c). This pocket is lined by Thr329, Leu332, Gly333, Ile336, Thr370, Ile411, Ile429, Glu433, Leu436, and Phe437, which are generally conserved in homologs (Supplementary Fig. 4e), supporting this structural motif's importance to binding the hydrophobic tail of S1P. Outside the pocket, the DDM's acyl chain makes further van der Waals contacts with TM1, TM5b, and TM10. Finally, the resolvable glucose moiety of the detergent makes a hydrogen bond with the conserved Trp440 of TM10 (Fig. 2c).

## Critical motifs to S1P export

Examining n-dodecyl-β-D-maltopyranoside's engagement with SPNS2, we noted that the detergent's head group interacts with the conventional MFS central cavity and is near a patch of conserved, polar residues on the N-domain (Fig. 2d, Supplementary Fig. 4f–h). Hypothesizing the DDM occupies the sphingosine-1-phosphate binding site, the conserved Arg119 of TM1b and His468 of TM11a adjacent to the detergent headgroup immediately suggested these residues coordinate the S1P's anionic head group. Supporting this notion, equivalent residues in SPNS1 and GlpT are essential to their transport of phosphate-containing substrates[32–34]. In contrast, the published state 1 and state 1* structures of SPNS2 in complex with S1P model the substrate head group interacting with TMs 5, 8, and 10[28]. While both molecules' tails similarly engage the transporter's pocket, the location of S1P's head group between TM5 and TM8 would interfere with a conventional MFS rocker-switch transport mechanism[29]. Therefore, we set out to experimentally probe the importance of several S1P interacting residues within SPNS2 to substrate export.

To probe each residue's role in the sphingosine-1-phosphate export activity of SPNS2, we first established an in vitro transport assay to report the export of substrate by taking advantage of the signaling cascade of the high-affinity S1P receptor 3 (Fig. 2e). Media was collected from CHO cells overexpressing sphingosine kinase 1 (SphK1) and SPNS2, or its mutants, after incubation with 1 µM sphingosine. Exported sphingosine-1-phosphate in the media was then quantified by the luminescence of reporter CHO cells expressing S1PR3 and mitochondrially-targeted obelin. Validating this assay, the SPNS2 mutation R200S shows no S1P transport activity (Fig. 2f), recapitulating the loss-of-function phenotype for the equivalent mutation in the *Danio rerio* ortholog[2]. Notably, this R200S mutant exhibits an increase in intracellular localization relative to the plasma membrane (Supplementary Fig. 5a). As Arg200 is on TM4 and makes intra-domain hydrogen bonds with the backbone of TM1's Asp118 and Arg119 (Supplementary Fig. 5b), this suggests misfolding of the N-domain likely impedes trafficking to the plasma membrane and thereby blocks the protein's S1P export activity.

As Gly333 has a dual role in sealing the extracellular gate of the inward-facing state and the acyl-chain binding pocket, we hypothesized that mutating this highly conserved side chain would significantly affect protein localization and activity. Supporting this hypothesis, the glycine-to-leucine mutation G333L reduces plasma membrane localization and sphingosine-1-phosphate export (Fig. 2f, Supplementary Fig. 5a). Curiously, changing the same position to phenylalanine in the mutant G333F further reduces plasma membrane

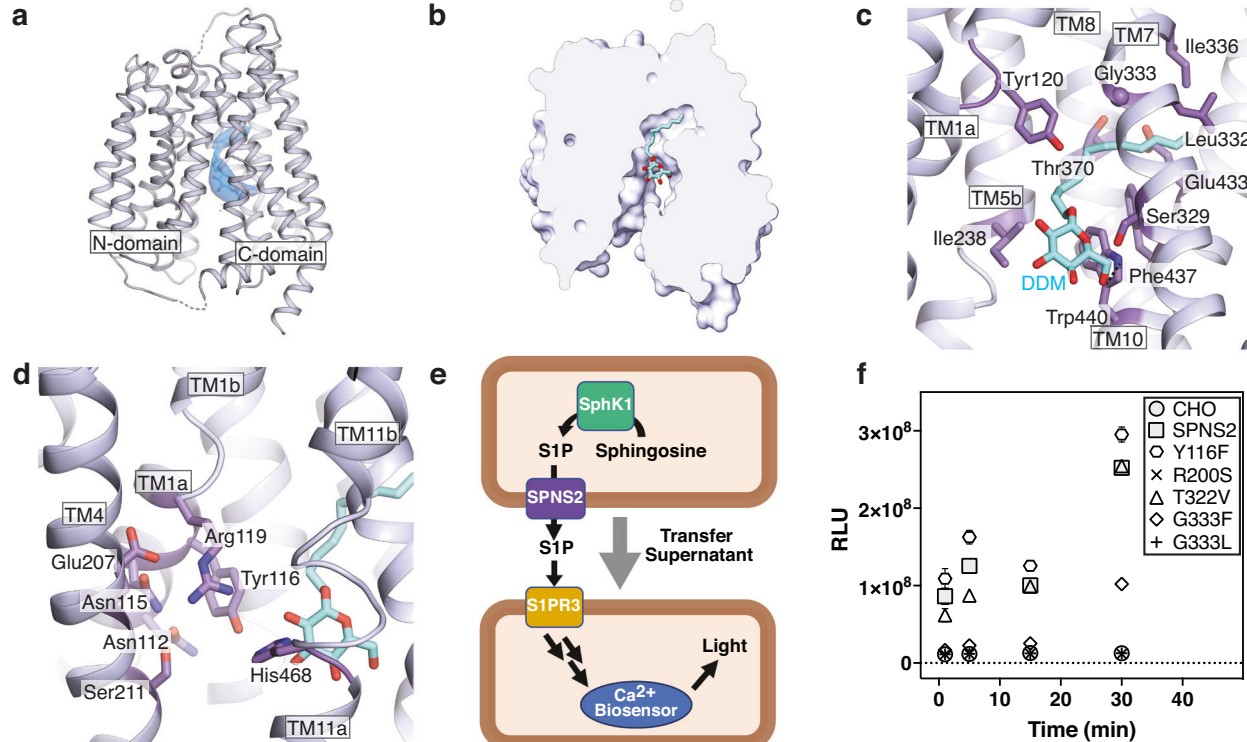

**Fig. 2 | DDM binding pocket of SPNS2. a** Bound n-dodecyl-β-D-maltopyranoside within the SPNS2 structure determined in DDM. Positive difference density of the weighted $F_o - F_c$ difference map at 16σ and the DDM model are shown as blue surface and DDM cyan sticks, respectively. **b** Cross section of the SPNS2 structure, viewed from the plane of the membrane. **c** SPNS2 coordinates DDM through van der Waals contacts in the pocket and hydrogen bonds in the central cavity. SPNS2-DDM hydrogen bonds are shown as dotted lines. **d** SPNS2 N-domain contains a patch of polar residues adjacent to the DDM head group. **e** Schematic for GPCR-based transport assay. Media from cells expressing SPNS2 and SphK1 is collected and applied to reporter cells expressing S1PR3 and the $Ca^{2+}$ biosensor obelin. **f** Measurement of S1P export activity by SPNS2 mutants (N = 24 independently treated samples). Data are presented as mean ± SEM.

localization while retaining partial transport activity. Nevertheless, the effect of mutating Gly333 on S1P export confirms the importance of this residue to SPNS2's transport cycle.

Next, we tested the importance of hydrophilic side chains within SPNS2's central cavity and likely near the substrate's head group. Both Tyr116 and Thr322 are highly conserved (Supplementary Fig. 2), and we expected these may hydrogen bond to S1P's phosphate, amine, or hydroxyl moieties. However, mutations to either position did not affect substrate export (Fig. 2f). This indicates both positions are individually dispensable for sphingosine-1-phosphate export, potentially by compensatory interactions of other residues within the central cavity of SPNS2. However, Y116F decreased plasma membrane localization while T322V increased protein on this membrane (Supplementary Fig. 5a), indicating these mutations may affect protein localization or folding.

**Substrate interactions with SPNS2**

As our mutagenesis experiments revealed that several highly conserved residues were dispensable for SPNS2's transport activity, we next applied atomistic molecular dynamics simulations to unambiguously resolve the transporter's interactions with sphingosine-1-phosphate and FTY720-P. Using the SPNS2-DDM structure's detergent as a guide for placing substrate, we performed 5 simulation replicates, each of duration 250 ns, starting with the substrate in the binding site.

In most simulations of SPNS2 with sphingosine-1-phosphate, the substrate is stably bound with its head group in the central cavity and the acyl chain within the C-domain's binding pocket (Supplementary Fig. 6a). Normalizing the frequency of contacts between substrate and transporter, we noted the acyl chain of S1P primarily interacts with the

pocket lining helices of TM7, TM9, and TM10 (Supplementary Fig. 6b). Further, by clustering all snapshots of the substrate during the simulations, we identified there were several major modes of substrate engagement by the transporter (Fig. 3a, Supplementary Fig. 6c, d). The most populous cluster has the head group engaging exclusively with the C-domain's Trp440 and Asp472 via polar interactions (Fig. 3a). In contrast, in the second and third most populous clusters the substrate makes bridging interactions between domains, connecting Arg119 and sometimes Tyr116 on the N-domain with His468 and Asp472 on the C-domain (Supplementary Fig. 6d). This supports the importance of the conserved polar patch of the N-domain to the transporter's substrate-triggered conformational change. Further, the mobility of the head group within the central cavity demonstrates multiple modes of SPNS2-head group interaction. This may explain the lack of an effect for the Y116F mutant on SPNS2 activity, as the nearby Arg119 can compensate in coordinating the S1P's phosphate moiety.

We next carried out the same simulation strategy with FTY720-P to probe how SPNS2 binds this unnatural substrate. As expected, interactions of SPNS2 with the acyl chains of S1P and FTY720-P were similar for both substrates (Supplementary Fig. 7b). However, the amine and hydroxyl moieties of FTY270-P interact less frequently with the C-domain relative to sphingosine-1-phosphate. Instead, these groups interact more often with the N-domain's polar patch residues Tyr116, Arg119, and Ile238. Further, we noted in the most populous clusters of simulations that polar moieties of the FTY720-P head group extensively interacted with the N-domain (Fig. 3b, Supplementary Fig. 7b, d), with fewer polar interactions to the C-domain relative to S1P. This appears to be a consequence of the bulky aromatic ring in FTY720-P sterically preventing these hydrogen bonds, though the unnatural substrate has several additional van der Waals contacts

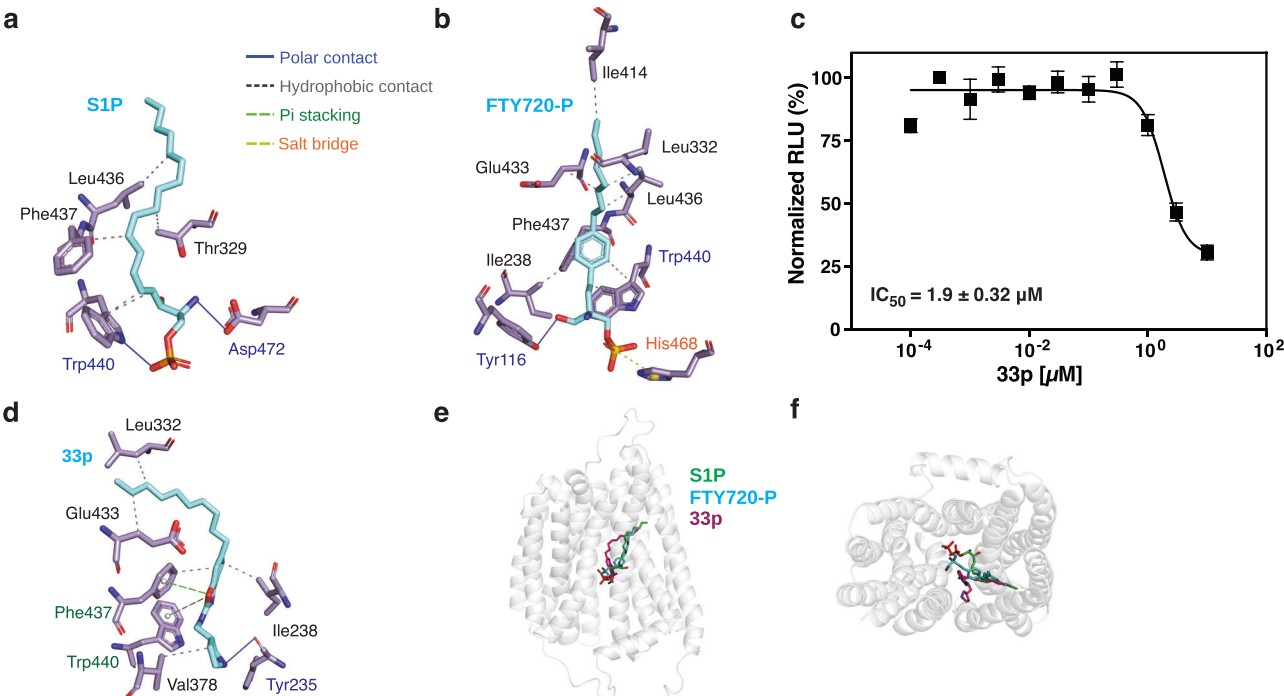

**Fig. 3 | Substrate and inhibitor binding by SPNS2. a** Sphingosine-1-phosphate head group interactions with SPNS2 in simulation cluster 1. Carbons of SPNS2 side chains and FTY720-P are shown in purple and cyan, respectively. Interaction types are annotated by PLIP[77]. **b** FTY720-P interactions with SPNS2 in simulation cluster 1. **c** Inhibition of S1P export by 33p measured by S1PR3-coupled export assay after incubation with 1 µM sphingosine (N = 8 independently treated samples). Data are presented as mean ± SEM. **d** 33p interactions within SPNS2 in simulation cluster 1. **e** Pose of cluster 1 for S1P, FTY720-P, and 33p with SPNS2 as viewed from the membrane plane. **f** Pose of cluster 1 for S1P, FTY720-P, and 33p with SPNS2 as viewed from the cytoplasm.

between its aromatic ring and the C-domain. Therefore, we propose hydrophobic interactions of FTY720-P with the C-domain substitute for the polar interactions by S1P to achieve similar substrate-transporter interaction energetics.

From these results of S1P and FTY720-P binding to SPNS2, we hypothesize the domain-bridging interactions of the substrates' phosphates pull together the N-domain and C-domain and thereby trigger the inward-facing to outward-facing conformational switch. This is analogous to the proposed transport of phosphate and glycerol-3-phosphate by GlpT[35].

## SPNS2 interactions with reported inhibitors

Building from these simulations of SPNS2 with substrates, we next sought to describe how the transporter is inhibited by the reported high-potency inhibitor 33p[20]. Applying it to the media of the SPNS2-expressing cells, we found the inhibitor had an IC₅₀ of 1.9 ± 0.32 µM in our S1PR3-coupled S1P export assay (Fig. 3c). The control compound, SphK1 inhibitor PF543[36], also inhibited S1P export with an IC₅₀ of 232 ± 55 nM (Supplementary Fig. 5c), while neither molecule significantly affected S1PR3 activity (Supplementary Fig. 5d). The IC₅₀ of 33p diminished with decreased external sphingosine (Supplementary Fig. 5e), suggesting the compound and S1P compete for a single binding site on SPNS2. However, this difference is not significant, likely due to this experiment's indirect manipulation of cytoplasmic S1P via varying concentrations of applied external sphingosine. This apparent inhibition of S1P export by 33p is 10-fold less potent than previously measured in HeLa cells by LC-MS/MS[20], and may reflect differences in the cell lines or experimental conditions used.

Examining the transporters' binding to 33p using atomistic molecular dynamics simulations, we observed the compound bound stably with its acyl chain inserted deeply into the C-domain pocket (Fig. 3d–f, Supplementary Fig. 8a). This is in contrast to the mobility of the low-potency 16d in a previous molecular dynamics run with

SPNS2[28], and we hypothesize the shallow modeling of this molecule into the transporter's C-domain pocket may affect its stability through the simulation. Clustering the 33p simulation snapshots revealed a skewed distribution (Supplementary Fig. 8c), indicating that the compound is biased to fewer conformations than substrates when bound to SPNS2. The secondary amines of 33p's head group make a hydrogen bond with the side chain hydroxyl of Tyr116 and pi stacking interactions with Trp440 (Fig. 3d, Supplementary Fig. 8d). Supporting the importance of this hydrogen bond to Tyr116, moving or removing the hydrogen bonding groups in analogs of 16d and 33p weakened their ability to interact with SPNS2[19,20]. However, altering either residue individually via W440A and Y116F mutations did not significantly alter the compound's inhibition of S1P export (Supplementary Fig. 8e). This suggests significant plasticity in the protein's binding to 33p, though we cannot exclude that the molecule might inhibit other proteins in the S1P synthesis and transport pathway.

Looking across all simulation snapshots, though also present in cluster 1, we noted 33p more frequently makes polar contacts to TM10 in the C-domain (Supplementary Fig. 8b) while making relatively fewer contacts with TM1. This immediately suggests a competitive inhibition mechanism for 33p, where the molecule binds with high affinity but cannot trigger a complete transport cycle. The numerous interactions of 33p with the C-domain suggest the molecule binds SPNS2 with high affinity. However, 33p's relatively few and rare interactions with TM1 of the N-domain are insufficient to trigger the transporter's inward-facing to outward-facing conformational change. Consequently, binding to 33p likely arrests SPNS2 in an inward-facing inhibitor-bound state.

## Apo structure of the transporter

To complement our detergent-bound structure of SPNS2, we next sought to determine the unseen inward-facing apo structure of SPNS2 and thereby describe the protein's conformational change. Examining the SPNS2-DDM structure, we hypothesize the C-domain's narrow acyl-

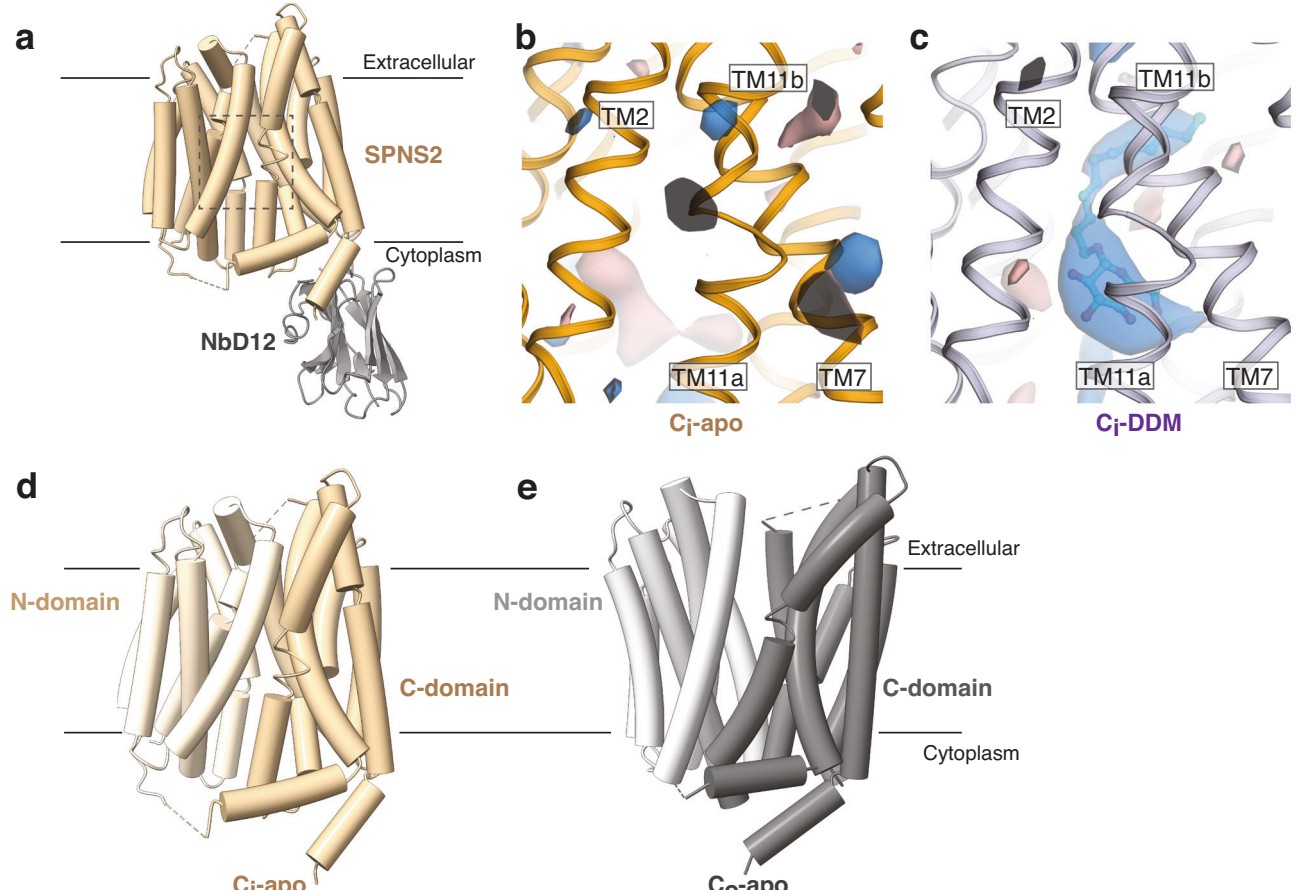

**Fig. 4 | Inward-facing apo structure of SPNS2. a** Structure of the SPNS2-NbD12 complex determined in LMNG, viewed from the plane of the membrane. The substrate and DDM binding site is indicated with a dotted box. Difference weighted $F_o$-$F_c$ density maps near the DDM binding site for the SPNS2-NbD12 complex cryo- EM maps determined in (**b**) LMNG and (**c**) DDM. Maps are contoured at equivalent levels using difference density from omitted sidechains to scale the difference maps. SPNS2 in (**d**) inward-facing and (**e**) outward-facing apo states (PDB: 8EX5).

chain binding pocket would sterically exclude larger detergents and therefore purified the protein in the larger diacyl detergent LMNG. NbD12-bound SPNS2 was monodisperse in LMNG (Supplementary Fig. 9a, b), and we determined the complex's structure in this detergent to a resolution of 3.7 Å by cryo-EM (Supplementary Fig. 9c, d).

Building and refining the SPNS2-NbD12 structure in LMNG (Fig. 4a), the proteins are generally unchanged from the DDM condition (all-atom RMSD = 0.47 Å). Notably, however, there is no apparent non-protein density in the central cavity or pocket of the SPNS2 map determined in LMNG (Supplementary Fig. 9e). Further, the difference between the observed Coulombic potential map and the predicted density from the protein-only SPNS2-NbD12 model gave a large positive density for the structure determined in DDM but not LMNG (Fig. 4b, c). This supports our assignment of the ambiguous density in the SPNS2-DDM map as n-dodecyl-β-D-maltopyranoside and indicates our LMNG map has captured the transporter's inward-facing apo ($C_i$-apo) state.

The most pronounced structural change in SPNS2 between apo and DDM-bound states is a ~90° rotation in Trp440, with its side chain rotating roughly parallel to the plane of the membrane in the absence of a head group for hydrogen bonding (Supplementary Fig. 10a-c). Further, while the density of Glu433's terminal carboxylate is weak, as is expected for this anionic moiety[37], this side chain rotates in the presence of DDM to accommodate the acyl chain and appears positioned to form a hydrogen bond with Thr373 (Supplementary Fig. 10d-f). The remaining residues of the substrate binding site do not significantly move, suggesting the substrate binding site

is largely pre-formed for efficient binding of amphipathic substrates (Supplementary Fig. 10g).

**Conformational change of apo SPNS2**

Comparing our inward-facing apo SPNS2 and the published outward-facing apo structures, the transporter has undergone a rigid body conformational change consistent with the classic MFS rocker-switch mechanism (Fig. 4d, e). In keeping with this mechanism, the individual N-terminal and C-domains are relatively unchanged between conformations, with all-atom RMSDs of 0.74 Å and 0.62 Å, respectively. However, the relative orientation of the domains has significantly changed due to an inter-domain rotation of 32°. Notably, no significant changes are observed at the NbD12 binding sites or at the termini where MBP and DARPin were fused in the previously published SPNS2$_{cryo}$ construct[28]. This structural consistency suggests the distinct methods of adding fiducial markers for particle alignment during cryo-EM reconstruction did not disturb the transporter's structure.

While SPNS2's rigid-body inter-domain movement accounts for most of the change between outward- and inward-facing conformations, we noted modest structural differences within the individual domains. The movement from $C_o$-apo to $C_i$-apo brings TM2 into greater contact with TM11, inducing an additional half-turn of alpha-helical structure in the previous pi helix (Supplementary Fig. 10h). This movement of TM2 also makes room for the movement of ICH1 toward the transport domain, with the angle between ICH1 and TM7 decreasing by 12° during the $C_o$-apo to $C_i$-apo transition (Supplementary Fig 10i). Relative to an idealized MFS rigid body movement, these

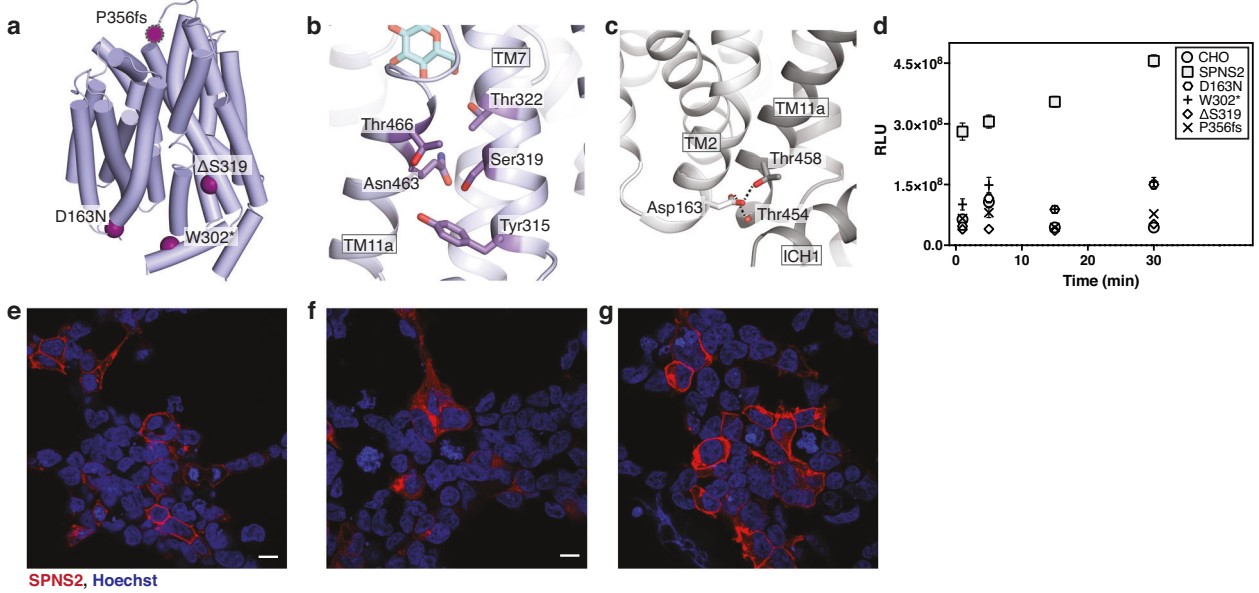

**Fig. 5 | Pathogenic mutations at essential locations in the SPNS2 structures.**
**a** Locations of pathogenic SPNS2 mutations within the structure. Loop 7-8 is not resolved in the structure, and the approximate location of Pro356 is indicated by a magenta sphere with a dotted edge. **b** The pathogenic mutation ΔS319 within the C-domain. **c** Asp163 in SPNS2 is within the conserved MFS motif A and forms hydrogen bonds specific to the outward-facing conformation (PDB: 8EX5). Hydrogen bonds are shown as dotted lines. **d** S1P export activity by pathogenic SPNS2 mutants (N = 24 independently treated samples). Data are presented as mean ± SEM. Anti-HA immunofluorescence for HEK293-JI cells transiently trans-fected with HA-tagged SPNS2 (**e**) wild-type or mutants (**f**) ΔS319 and (**g**) D163N. Representative image with clear signal of the relevant immunofluorescence chan-nel and appropriate Z plane after collecting 3-5 images from different fields of biological duplicates.

conformation-specific structural changes likely bias the energetics of SPNS2's outward-facing to inward-facing states.

## Mechanism of pathogenic SPNS2 mutation in deafness

With our structures of SPNS2 and a model for its S1P transport mechanism, we next sought to understand the pathogenic mechanism for four clinical mutations in SPNS2 that cause hearing loss (Fig. 5a)[12,13]. Of these, the nonsense W302* mutation introduces a premature ter-mination at the start of ICH1. Similarly, the frameshift mutation P356fs, located in the loop between TM7 and TM8, leads to mistranslation and subsequent premature stop codon. Ultimately, these incomplete proteins lack significant portions of the C-domain and, therefore, are likely incapable of transport, and accordingly both mutations had insignificant, or dramatically reduced, S1P export (Fig. 5d).

In contrast to the pathogenic effects of premature truncations, the pathogenic SPNS2 mutations ΔS319 and D163N are associated with hearing loss yet produce full-length or near-full-length proteins. Examining its location within the transporter's structure, we noted Ser319 lies on TM7 and within a conserved set of hydrophilic residues packed between that helix and TM11a (Fig. 5b). We hypothesize that deletion of this residue causes a register shift of that helix, which would result in aberrant packing of TM7 within the C-domain. Supporting this hypothesis, the ΔS319 failed to traffic to the plasma membrane when transiently transfected into HEK cells (Fig. 5f) and resulted in the loss of activity for the mutant (Fig. 5d).

A third mechanism appears to cause the loss in activity of the pathogenic SPNS2 mutant D163N, as it is on the surface of the N-domain and, therefore, unlikely to affect protein translation or folding (Fig. 5a). Notably, Asp163 is within the conserved motif A of MFS transporters implicated in transporter gating[38]. Accordingly, we noted that while the side chain points toward the cytoplasm in our inward-facing state structures, it interacts with a bend in TM11 of SPNS2's outward-open and outward-occluded states (Fig. 5c). Here, it forms hydrogen bonds with the side chain of Thr458 and the backbone

carbonyl of Thr454. Therefore, we expect the D163N mutation to weaken these interactions, disturbing the energetics of SPNS2's transport cycle without significantly affecting protein folding. Sup-porting this hypothesis, the D163N mutant of SPNS2 correctly traffics to the plasma membrane (Fig. 5g) and has partial S1P export activity (Fig. 5d). Based on the modest activity of the D163N mutant, we hypothesize there is a minimal S1P export activity needed from SPNS2 for proper development of the stria vascularis, below which it becomes disorganized and thereby leads to hearing loss.

## Discussion

In this study, we identify the biophysical mechanisms of SPNS2's interactions with sphingosine-1-phosphate, the immunomodulator FTY720-P, and the reported inhibitor 33p through a combination of structural studies, in vitro binding, in vivo transport, and molecular dynamics simulations. These results provide a framework for under-standing the transporter's mechanism for substrate export, and interaction with SPNS2-targeting small molecules. Further, by captur-ing an inward-facing apo state, we identify modest changes in the transporter's secondary structure which affect the outward-to-inward conformational change. Finally, coupling our structural results and protein localization, our results explain the pathogenic effects of SPNS2 mutations implicated in hearing loss.

Notably, our analysis of SPNS2 binding to substrates S1P and FTY720-P provides insights into the transporter's action on those molecules. In our simulations, we did not observe S1P or FTY720-P binding between TM5 and TM8, as was modeled previously for S1P[28]. It remains unclear if the previously modeled S1P location is an inter-mediate step of the reaction cycle, an artifact of the substrate inter-acting with the saposin nanodisc, or a modeling error due to poor local resolution. Consistent with S1P's head group mobility we observed by molecular dynamics, in S1P-bound SPNS2 structures released while under review the headgroup is modeled in several locations while the tail is consistently inserted into the C-domain pocket of the relatively

rigid inward-facing transporter (RMSD = 0.55–0.97 Å) (Supplementary Fig. 10j)[39,40]. Nevertheless, our updated location for substrate binding supports SPNS2 using a classic MFS rocker-switch transport mechanism. Further, our structural and MD analysis identified conserved residues within the N-domain's polar patch that are likely critical to triggering the conformational change, supported by the failure of the S1P export inhibitor 33p to engage these side chains. This makes clear that engaging both domains is central to triggering the transport cycle, and is highly analogous to the phosphate bridging seen in GlpT[35]. Further, the binding mode of SPNS2 to substrate is distinct from other S1P-binding proteins and therefore may have significant implications for the molecule's binding kinetics and affinity. In particular, the S1P head group is exposed to solvent in SPNS2 and ApoM[41], but inaccessible to solvent in S1PR1 and S1PR3[42–44]. Additionally, the acyl chain of S1P-bound ApoM, and sphingosine bound SphK1[45], adopts a J-shape while it is extended in the other proteins. Ultimately, these differences in substrate engagement and geometry may aid the design of small molecules that specifically target SPNS2.

These structures also hint at a role for SPNS2's Glu433, which is unusual for being a charged residue conserved within the acyl-chain binding pocket and the hydrophobic core of the C-domain. Examining all available structures, we noted this side chain is highly mobile within the pocket in the absence of S1P or analogs and potentially interacts with solvent or Thr370 (Supplementary Fig. 10d). However, in available structures of SPNS2 with loaded substrate or inhibitor, Glu433 consistently makes a hydrogen bond with Thr373. This same Thr373-Glu433 hydrogen bond is also found in the outward-occluded apo state structures of SPNS2[28]. Therefore, it is tempting to hypothesize the orientation and hydrogen bonding of Glu433 is acting as a switch to enable the transporter's conformational change which triggered by the substrate's acyl chain, or stochastically in the apo state. How the state of this glutamate is propagated to affect more significant structural changes in the protein is unclear. However, we suspect in an extended conformation Glu433 could hydrogen bond more frequently or strongly with Thr370 and nearby waters, thereby altering the outward-facing state's energetic stability.

Notably, in our molecular dynamics study, one simulation each for sphingosine-1-phosphate and FTY270-P showed the substrate leaving the binding site and moving closer to the cytoplasmic surface of the protein (Supplementary Fig. 6a, 7a). These transits of substrate away from the binding site are too rare for statistical analysis. Nevertheless, the S1P and FTY720-P travel we observed align with the orientation of these molecules in the inner membrane after synthesis inside the cell, and hint at a path for substrate movement to and from the transporter's binding site.

Finally, while SPNS2's essential role in the pharmacokinetics of fingolimod by exporting FTY720-P is well known, our structural and biochemical results suggest the transporter may play a role in the metabolism and movement of other therapeutic small molecules. Our DDM-bound structure indicates SPNS2 is relatively promiscuous for amphiphilic molecules with a single acyl chain. Therefore, SPNS2 may play a role in the pharmacokinetics of other analogs of sphingosine-1-phosphate which target the S1P receptors, and other pharmacophores with a single acyl chain.

## Methods

### Ethical statement

The immunizations of alpaca were conducted strictly according to the guidelines of the Swiss Animals Protection Law and were approved by the Cantonal Veterinary Office of Zurich, Switzerland (Licenses No. ZH 198/17 and ZH028/2021).

### Sequence alignment

SPNS2 orthologs, MFS-fold lipid transporters, and prototype MFS transporters were aligned in Promals3D[46] and rendered with ESPript[47].

### Cloning, expression, and purification of SPNS2

The full-length human SPNS2 gene was cloned into the pHTBV1.1 plasmid containing a C-terminal tobacco etch virus (TEV) protease cleavage site followed by EGFP, twin-Strep, and 10×His affinity tags. Baculovirus was then generated according to the previously described protocols[31,48,49].

The resulting baculovirus was used to infect Expi293F cells in Freestyle 293 expression medium (GIBCO) in the presence of 5 mM sodium butyrate. Infected cells were grown in an orbital shaker for 72 h at 37 °C, 8% $CO_2$ and 75% humidity, harvested by centrifugation, washed with phosphate-buffered saline, flash-frozen, and stored at -80 °C until further use.

The cell pellets were resuspended in extraction buffer (300 mM NaCl, 50 mM HEPES pH 7.5, 1% DDM or 1% LMNG) in the presence of cOmplete Protease Inhibitor Cocktail tablets (Roche) and solubilized at 4 °C for 1 h with gentle rotation. The insoluble materials were pelleted at 50,000 × g for 40 min. The supernatants were incubated with pre-equilibrated TALON resin (Takara) and allowed to bind for 1 h at 4 °C. The resin was poured onto a gravity-flow column and washed with column buffer (300 mM NaCl, 50 mM HEPES pH 7.5, and 0.03% DDM (Anatrace) or 0.01% LMNG (Anatrace)) supplemented with 10 mM MgCl₂, 1 mM ATP, and 10 mM imidazole. Protein was eluted with elution buffer (300 mM NaCl, 50 mM HEPES pH 7.5, 300 mM imidazole and 0.03% DDM or 0.01% LMNG). The eluate was incubated with pre-equilibrated Strep-Tactin XT Superflow resin (IBA-Lifesciences) for 1 h at 4 °C. The resin was poured onto a gravity-flow column and washed with column buffer. Protein was eluted with column buffer supplemented with 50 mM D-biotin, followed by tag-cleavage with TEV protease overnight and reverse IMAC purification using TALON resin.

The tag-cleaved SPNS2 proteins were concentrated using a centrifugal concentrator with 100 kDa cut off (Sartorius) and subjected to size exclusion chromatography using a Superdex 200 10/300 GL column (GE Healthcare) pre-equilibrated with gel filtration buffer (150 mM NaCl, 20 mM HEPES pH 7.5, 0.025% DDM or 0.002% LMNG). Peak fractions were pooled and concentrated for subsequent experiments.

### Cloning, expression, and purification of biotinylated SPNS2

Full-length codon-optimized SPNS2 with a C-terminal AVI and Flag tag and BirA were cloned into in-house Baculovirus expression vector pD-INS3, and baculovirus produced using standard methods[48,49]. For protein expression, Sf9 cells at $4 \times 10^6$ cells/mL were co-infected with SPNS2 and BirA virus with MOI 1 and 0.3, respectively, and cultured at 27 °C with biotin supplemented in the medium. Cells were harvested by centrifugation after 72 h, and cell pellets solubilized in 50 mM HEPES pH7.5, 300 mM NaCl, 10% glycerol supplemented with protease inhibitor, benzonase, 2 mM biotin and 1% DDM or 1% LMNG. After solubilization at 4 °C for 1.5 h, insoluble material was removed by ultracentrifugation at 35,000 × g for 1 h. Supernatant was incubated with Anti-Flag M2 resin (Sigma) for 1.5 h at 4 °C. The resin was washed with 30 column volumes of 50 mM HEPES pH 7.5, 300 mM NaCl, 5% glycerol with 0.026% DDM or 0.01% LMNG. The protein was eluted with 50 mM HEPES pH 7.5, 300 mM NaCl, 5% glycerol, 200 µg/ml Flag-peptide with 0.026% DDM or 0.01% LMNG. Eluted protein fractions were concentrated using a 100 kDa Amicon Ultra centrifugal filter (Millipore) and applied to a Superose 6 increase 10/30 GL (GE) gel filtration chromatography column pre-equilibrated in 20 mM HEPES pH 7.5, 150 mM NaCl with 0.026% DDM or 0.003% LMNG. SEC fractions were analyzed by SDS-PAGE and pooled accordingly. Purified SPNS2 was supplemented with 10% glycerol, aliquoted, flash frozen and stored at -80 °C until use.

### Cloning, expression, and purification of VNUP

Full-length VNUP was cloned into in-house mammalian expression vector pD-MAM8.1 with a C-terminal AVI and Flag tag. For protein

expression, Expi293F cells at $1 \times 10^6$ cells/mL were transiently transfected with 1 mg/L DNA/PEI complex with DNA:PEI ratio of 1:3. Cells were incubated at 37 °C, 8% $CO_2$ for 72 h post-transfection and harvested by centrifugation. Cell pellets were solubilized in 50 mM Tris-HCl pH 7.5, 200 mM NaCl, 20 mM KCl, 30% glycerol supplement with protease inhibitor, benzoase and 1% LMNG (Anatrace). After solubilization at 4 °C for 1.5 h, insoluble material was removed by ultracentrifugation at $35,000 \times g$ for 1 h. Supernatant was incubated with Anti-Flag M2 resin (Sigma) for 1.5 h at 4 °C. The resin was washed with 20 column volumes of 50 mM Tris-HCl pH 7.5, 100 mM NaCl, 10 mM KCl, 1 mM ATP, 10 mM $MgCl_2$, 20% glycerol, 0.01% LMNG, followed by a second wash step with 50 mM Tris-HCl pH 7.5, 100 mM NaCl, 10 mM KCl, 20% glycerol, 0.01% LMNG. The protein was eluted with 50 mM Tris-HCl pH 7.5, 100 mM NaCl, 10 mM KCl, 20% Glycerol, 0.01% LMNG, and 200 µg/ml Flag-Peptide. Eluted protein fractions were concentrated using a 50-kDa molecular-weight (MW) cut-off Amicon Ultra centrifugal filter (Millipore), and further purified using size exclusion chromatography on a Superose 6 Increase 10/30 GL in 50 mM Tris-HCl pH 7.5, 100 mM NaCl, 10 mM KCl, 10% Glycerol, 0.01% LMNG. Peak fractions were analyzed by SDS-PAGE and pooled accordingly. Purified VNUP was aliquoted, flash frozen, and stored at −80 °C until use.

### Thermal stability assay of SPNS2 and VNUP
The thermal stability assay was performed with nanoDSF assay. SPNS2 or VNUP at 1 µM concentration were incubated with 0–160 µM of FTY720-P (stock concentration 1.6 mM in DMSO) in 20 mM HEPES pH 7.5, 150 mM NaCl, 10% DMSO with 0.026% DDM or 0.003% LMNG. All samples were incubated for 30 min at room temperature prior to analysis. After incubation, samples were loaded into high sensitivity grade nanoDSF capillaries (NanoTemper), measured in Prometheus NT.48 device (NanoTemper) with excitation power 100% and temperature gradient from 20 °C to 90 °C with a slope of 2 °C/min. Data were analyzed using PR ThermControl software (NanoTemper).

### Nanobody generation
To induce the development of heavy chain-only antibodies (IgG2 and IgG3) in alpacas, animals were immunized four times at 2-week intervals, each time with 150–200 µg of purified protein. All the procedures concerning alpaca immunization were approved by the Cantonal Veterinary Office of Zurich, Switzerland (License No. ZH 198/17). SPNS2 was delivered in proteoliposomes consisting of soy asolectin, porcine brain polar lipid extract, cholesterol, and monophosphoryl hexa-acyl lipid A (Avanti Polar Lipids) at a ratio of 24:8:7:0.5 by weight in PBS. Before injections, antigens were mixed in a 1:1 (vol/vol) ratio with GERBU Fama adjuvant (GERBU Biotechnik GmbH, Heidelberg, Germany) and injected subcutaneously in 100 µL aliquots into the shoulder and neck region. Two weeks after the last boost, 80 mL of blood was collected from the jugular vein for isolation of lymphocytes (Ficoll-Paque PLUS, GE Healthcare Life Sciences, and Leucosep tubes, Greiner). Approximately 50 million cells were used to isolate mRNA (RNeasy Mini Kit, Qiagen) that was then reverse transcribed into cDNA (AffinityScript, Agilent, USA) using the VH gene-specific primer. The $V_H H$ (nanobody) repertoire was amplified by two PCRs and phage library was generated using established methods[50], fragment exchange cloning into a SapI-linearized pDX phagemid vector using 336 ng of the $V_H H$ repertoire and 1 µg of the plasmid DNA.

The resulting nanobody library (size $2 \times 10^8$) was screened by biopanning against indirectly immobilized targets. For this purpose, biotinylated SPNS2 in 20 mM HEPES pH 7.5, 150 mM NaCl with 0.015% DDM and 0.0015% CHS was immobilized (1 µg SPNS2 per well) on Streptavidin or Neutravidin-coated microplates (alternating between selection rounds) at 5 µg/mL, 100 µL per well in 96 well Maxisorp plate (Nunc, Denmark) and two rounds of selections were performed until ~1000-fold positive enrichment of phages was obtained. Single clones for 190 nanobodies were expressed as polyhistidine-tagged soluble nanobodies in the bacterial periplasm and analyzed by ELISA for binding to SPNS2.

Ninety-six ELISA-positive clones were Sanger sequenced and a phylogenetic analysis with the resulting binder sequences for each target was performed using MMseqs2[51]. Sequences were clustered based on a minimum sequence identity of 85% and representative sequences were chosen from 9 clusters. Selected binder sequences were subcloned into a pBXNP plasmid (Addgene #110098) that was modified to contain 1x FLAG tag upstream of the 10x His tag. Binders were finally purified from the periplasm of MC1061 bacteria according to the published protocol[52].

### Surface plasmon resonance assay of SPNS2
The binding affinities of Nanobody D12 and F09 were determined using Biacore 8 K machine (Cytiva). Biotinylated SPNS2 protein was immobilized on the SA sensor chip to a target level of 600 to 800 RU. Purified nanobodies were serial diluted with running buffer (20 mM HEPES pH 7.5, 150 mM NaCl, 0.026% DDM, 0.01% fatty acid free BSA) to a concentration gradient between 0 to 1 or 10 µM. The diluted nanobody series were injected over the immobilized sensor in running buffer at 25 °C while binding traces monitored simultaneously. Data fitting was analyzed with 1:1 interaction model using the Biacore Insight Evaluation software (Cytiva).

### Cell lines and culture conditions
The Jump In T-REx human embryonic kidney 293 cell line (HEK293-JI) expressing doxycycline-inducible human SPNS2 were generated by RESOLUTE as described previously[53]. Briefly, codon-optimized sequences of wild type and mutant SPNS2 in pDONR221 (Addgene #132307) were subcloned into a modified pJTI R4 DEST CMV TO pA plasmid (Thermo Fisher Scientific) containing Twin-Strep-Tag (IBA Lifesciences) and HA epitopes (SH tag) at the N or the C terminus of SPNS2 as indicated. All constructs were confirmed by Sanger sequencing. HEK293-JI cells stably expressing each construct were selected in DMEM medium (Sigma-Aldrich, D5796) supplemented with 10% FBS and 5% Pen/Strep (Sigma-Aldrich, P4333), 0,0005% Blasticidin (Invivogen, ant-bl-05) and 4% Geneticin (Sigma-Aldrich, A1720) for one week. After selection, cells were kept in DMEM medium supplemented with 10% FBS and 5% Pen/Strep and protein expression was induced for 24 h with 10 µg/mL doxycycline (Sigma-Aldrich, D9891).

### Immunofluorescence
To detect localization of SPNS2 in HEK293-JI-SPNS2-SH cells, the cells were seeded at a density of $1.2 \times 10^5$ on glass microscopic cover slips coated with Poly-L-lysine (Sigma P2658) already placed into 24-well plates (Corning). 16 h after protein induction with 1 µg/mL doxycycline, cells were fixed and permeabilized in 4% PFA for 15 min at RT. Cells were subsequently incubated in blocking buffer containing 10% FCS and 0.3% Triton X-100 in 1x PBS for 1 h at room temperature. Next, samples were incubated with anti-HA from rat (Roche, 60789700, 1:1000) antibody for 2 h at room temperature. After three washing steps, cells were incubated with secondary antibodies anti-rat coupled to Alexa Fluor 488 (Thermo Fisher Scientific, A-11006, 1:500) and DAPI (Sigma, D9542, 10 mM) diluted 1:1000 for 1 h at room temperature. After three washing steps (two times blocking buffer, one-time 1x PBS), the slides we mounted with ProLong medium diamond (Invitrogen) and stored at 4 °C. Imaging was performed using confocal microscope LSM980 (Zeiss) with 63x objective and obtained images were processed in Zen blue 3.3 software (Zeiss).

To detect co-localization of NbD12 and NbF09 with SPNS2 and localization of SPNS2 transport mutants (using NbF09) the protocol above was used with following the adaptations: after fixation and blocking, the cells were firstly incubated with NbD12 or NbF09 for 2 h at room temperature in presence of 0.3% TRITON-X 100. After three washing steps, samples were incubated simultaneously with anti-HA

from rabbit (Cell Signaling Technology, C29F4, 1:400) and anti-FLAG-M2 from mouse (Sigma, F3165, 1:1000) antibodies for 2 h at room temperature. After three washing steps, cells were simultaneously incubated with secondary antibodies goat anti-rabbit coupled to Alexa Fluor 594 (Thermo Fisher Scientific, A-11012, 1:500), goat anti-mouse coupled to Alexa Fluor 488 (Thermo Fisher Scientific, A-11001, 1:500) and DAPI (Sigma D9542 10 mM) diluted 1:1000 for 1 h at room temperature.

To probe localization of SPNS2 hearing loss mutants ΔS319 and D163N, the mutations were introduced by site-directed mutagenesis into the pDONR221-SPNS2_STOP plasmid (Addgene #161473)[54]. WT and mutant SPNS2 cDNAs were subsequently cloned into a modified pJTI R4 CMV-TO MCS pA vector (Thermo Fisher Scientific) that contained an N-terminal HA-Twin-Strep tag (see Addgene plasmid #194065 for tag sequence). All plasmids were validated by Sanger sequencing. Jump-In T-REx HEK293 cells (Thermo Fisher Scientific) were stably transfected with these constructs as recommended by the manufacturers protocol. Transgene expression was induced via doxycycline treatment (1 μg/ml) for 24 hours before analysis. The immunofluorescence protocol above was performed with the following adaptations: after fixation and blocking, the cells were incubated with anti-HA from rabbit (Cell Signaling Technology, C29F4, 1:400) and simultaneously incubated with secondary antibody goat anti-rabbit coupled to Alexa Fluor 594 (Thermo Fisher Scientific, A-11012, 1:500) and DAPI (Sigma D9542 10 mM) diluted 1:1000 for 1 h at room temperature.

### Cell lysis, co-immunoprecipitation, and western blotting
Approximately 10 x 10⁶ cells were lysed in 250 μL of lysis buffer composed of 50 mM HEPES pH 7.4, 250 mM NaCl, 5 mM EDTA, and 1% NP-40, supplemented with Roche EDTA-free protease inhibitor cocktail (1 tablet per 50 mL) and incubated for 30 min on ice. Lysates were cleared by centrifugation at 13,000 rpm, 15 min, 4 °C and total protein concentration was quantified using a Bradford protein assay (Bio-Rad). Samples were diluted to 0.5 mg of total protein per sample. Subsequently, 200 μL clarified and diluted lysate was mixed with 1 μg nanobody and incubated at 4 °C overnight. Immunoprecipitation was carried out using equilibrated anti-FLAG M2 affinity gel (Sigma, #A2220) for 2 h at 4 °C, with beads collected by centrifugation at 13,000 rpm, 15 min, 4 °C and subsequently washed three times with 1x TBS buffer. The bound protein fraction was eluted with 0.1 M glycine-HCl. For western blot analysis, 1 μL of input and 3 μL of each eluted sample were run on 10% SDS-polyacrylamide gel in Tris-Glycine running buffer and transferred to nitrocellulose membranes Amersham Protran 0.45 mm (GE Healthcare). The membranes were blocked with 5% non-fat dry milk in TBST and probed with primary antibodies anti-HA from rabbit (Cell Signaling Technology, C29F4, 1:2000) and anti-FLAG-M2 from mouse (Sigma, F3165, 1:2000) at 4 °C overnight followed by secondary antibodies goat anti-mouse HRP (115-035-003, Jackson ImmunoResearch) and goat anti-rabbit HRP (111-035-003, Jackson ImmunoResearch). Binding was detected with horseradish-peroxidase-conjugated secondary antibodies using the ECL western blotting system (Thermo Fisher Scientific).

### Cryo-EM sample preparation and data collection
For purification of the SPNS2-NbD12 complex, SPNS2 after reverse IMAC purification and SEC purified NbD12 were mixed at 1:1.5 molar ratio, incubated at 4 °C for 1 h, and the complex purified by size exclusion chromatography using a Superdex 200 10/300 GL column (GE Healthcare) pre-equilibrated with gel filtration buffer (150 mM NaCl, 20 mM HEPES pH 7.5, 0.025% DDM or 0.002% LMNG).

Cryo-EM grids of SPNS2-NbD12 were prepared on freshly glow discharged QuantiFoil Au R1.2/1.3 300-mesh grids (Quantifoil) using a Mark IV Vitrobot (Thermo Fisher Scientific) at 100% humidity and 4 °C, and then plunged into liquid ethane. Peak fractions of SPNS2-NbD12 in DDM were pooled, concentrated to 14 mg/mL, and plunge-frozen on the QuantiFoil grid after blotting for 5.0 s. Peak fractions of SPNS2-NbD12 in LMNG were similarly pooled and concentrated to 6 mg/mL, then plunge-frozen on the QuantiFoil grid after blotting for 8.0 s.

The SPNS2-NbD12 in DDM dataset was collected on a Titan Krios electron microscope, using a GIF-Quantum energy filter with a 20 eV slit width (Gatan) and a K3 direct electron detector (Gatan) at a magnification of 105000x for a pixel size of 0.829 Å. EPU (Thermo Fisher Scientific) was used to automatically record three movie stacks per hole in super-resolution mode with 2x binning with the defocus ranging from −1.0 to −2.4 μm. At a dose rate of 17.5 e⁻/px/sec, each micrograph was dose-fractioned into 40 frames, for an accumulated dose of 40.74 e⁻/Å².

The SPNS2-NbD12 in LMNG dataset was collected on a Titan Krios electron microscope, using a GIF-Quantum energy filter with a 20 eV slit width (Gatan) and a K3 direct electron detector (Gatan) at a magnification of 105000x for a pixel size of 0.831 Å. EPU (Thermo Fisher Scientific) was used to automatically record two movie stacks per hole in super-resolution mode with 2x binning, at a dose rate of 14.3 e⁻/px/sec with the defocus ranging from −1.0 to −2.4 μm. Each micrograph was dose-fractioned into 40 frames, with an accumulated dose of 41.42 e⁻/Å².

### Reconstruction of SPNS2-NbD12
cryoSPARC was used for both data processing workflows[55]. Movies were patch motion corrected and CTF-corrected and manually curated based on ice thickness and CTF fit resolution.

For the DDM dataset, particles were blob-picked, followed by two cycles of 2D classification. The particles from well-resolved 2D classes were used for template-based picking and the resultant particles were then subjected to 2D classification, ab initio models generation and preliminary heterogeneous classifications. Particles from a good class of heterogeneous refinement were used for Topaz[56] training and picking. Topaz and template-picked particles were then combined and duplicates removed. Poorly aligning particles were removed by two cycles of 2D classification followed by iterative ab initio 3D model generation and heterogeneous classifications. Non-uniform refinement yielded a reconstruction for the SPNS2-NbD12 complex at 3.68 Å. To improve the map, local refinement was performed with the mask applied to exclude the DDM micelle and yielded a final map at a nominal resolution of 3.68 Å.

For the LMNG dataset, particles were blob picked, followed by three cycles of 2D classification. The particles from well-resolved 2D classes were used for template-based picking followed by further 2D and 3D classification. Particles from the best resolved class of heterogeneous refinement were then used for Topaz training and picking. Topaz and template-picked particles were then combined, and duplicates removed. Poorly aligning particles were then removed through iterative ab initio 3D model generation. Non-uniform refinement yielded reconstructions for the SPNS2-NbD12 complex at 3.98 Å. Local refinement was performed with the protein-only mask applied and yielded a map at 3.69 Å.

### Model building and refinement
Models were initially built using the DDM dataset. The human SPNS2 protein model from AlphaFold[57] and the nanobody protein model from published structures of PCFT[58] were roughly fitted into the experimental map and used as templates for model building in Coot[59]. Following manual adjustments, models were refined with phenix.real_space_refine using default geometric restraints[60]. For the LMNG dataset, the SPNS2-NbD12 model in DDM was used as template in Coot. Only the acyl chain and first glucose unit of the DDM headgroup were well resolved in the SPNS2-DDM cryo-EM map and have

been included in the model. Residual density for the second glucose unit is present at lower contour level but is somewhat disconnected from the rest of the molecule and was not included in the final model. Secondary structure within the models was identified by DSSP[61]. Model-to-map FSCs and validation statistics are listed in Supplementary Table 1.

Weighted $F_o$-$F_c$ difference maps used to highlight non-protein features are calculated automatically as part of the refinement pipeline using ServalCat in the CCPEM software package[62]. Briefly, all ligands were removed from each final SPNS2 model and then refined using ServalCat against its cryo-EM map for twenty cycles with jelly-body restraints to generate the difference maps. For visualization purposes, the difference maps for the DDM and LMNG datasets were scaled relative to one another by using the $F_o$-$F_c$ peak difference heights corresponding to well-defined Tyr side chains near the binding site of each model.

## S1PR3-coupled transport assay

Generation and cloning of SPNS2 wild type and mutant sequences, without any tags, into the piggyBac vector was performed by DNA-Cloning-Service e.K. (Hamburg, Germany). SphK1 and SPNS2 wild type or mutant overexpressing cells were generated by transfection of CHO-K1 cells using GeneJET according to manufacturer´s specification (Roche Diagnostics GmbH, Mannheim, Germany). Stably transfected clone pools were obtained by selection with 0.25 mg/mL Zeocin (InvivoGen, San Diego, CA, USA). CHO cells stably overexpressing S1PR3 and mitochondrially-targeted obelin were generated in-house (Bayer AG, Wuppertal, Germany). All cell lines were cultured in PAN medium (PAN Biotech, Aldersbach, Germany) containing 10% dialyzed FCS and kept under sterile conditions at 37 °C and 5% $CO_2$.

All luminescent measurements of the activation assay were performed on 384-well microtiter plates (MTPs) using a FLIPR Tetra High-Throughput Cellular Screening System (Molecular Devices, San Jose, CA, USA). Therefore, 5000 cells/well of SPNS2 wild type and mutant transfected cells were seeded in 25 µL Optimem containing 1% dialyzed FCS. For S1PR3 transfected cells, 5000 cells/well were seeded in 20 µL Optimem containing 1% dialyzed FCS with 5 µg/mL coelenterazine. Cells were incubated for 24 h at 30 °C and 5% $CO_2$. Sphingosine (Sigma-Aldrich, Munich, Germany) dilution was prepared in Tyrode (130 mM NaCl, 5 mM KCl, 20 mM HEPES, 1 mM $MgCl_2$, 2 mM $CaCl_2$, 4.8 mM $NaHCO_3$ at pH 7.4) containing 0.3% BSA and added to the SPNS2 wild type and mutant cells with a final concentration of 1 µM. Cells were incubated for 1, 5, 15 or 30 min at 37 °C and 5% $CO_2$. In the following step, 20 µL supernatant from SPNS2 wild type and mutant cells was transferred to the S1PR3 cells within the FLIPR device and luminescence signals, expressed as relative light units (RLUs), were measured for 60 sec.

To evaluate the effects of PF543 and the reported SPNS2 inhibitor 33p (SLB1122168), IUPAC name 6-decyl-N-[[(3S)-pyrrolidin-3-yl]methyl]-1,3-benzoxazol-2-amine, SPNS2 wild type and S1PR3 overexpressing cells were seeded as described. A dilution series of 33p (Probechem Biochemicals Co Ltd, Shanghai, China) and PF543 (Sigma-Aldrich, Munich, Germany) was performed in DMSO and Tyrode containing 1:500 SmartBlock (Candor Bioscience GmbH, Wangen, Germany) was added. The dilution series, starting at 10 µM, was then transferred to the SPNS2 cells and incubated for 10 min at room temperature. Subsequently, sphingosine was added to the SPNS2 cells and incubated for a further 15 min at 37 °C and 5% $CO_2$. Supernatant of the SPNS2 cells was then transferred to the S1PR3 cells within the FLIPR device and luminescence signals were measured for 60 sec. The effect of 33p and PF543 on the S1PR3 was quantified by adding the compounds' dilution series, starting at 10 µM, onto the S1PR3 overexpressing cells and incubating for 10 min at room temperature. Subsequently, the luminescence signals were measured for 60 sec in the FLIPR device.

## Molecular dynamics simulations

The coordinates of the inward-facing conformation of SPNS2 were obtained from its DDM-bound cryo-EM structure. All missing residues of SPNS2 and its residues with unresolved sidechains were modeled using Modeler 9[63]. The topologies for all substrates i.e., S1P, FTY720-P, and 33p, were parametrized using the CHARMM-GUI ligand reader and modeler[64] and the CHARMM36 general forcefield[65]. The phosphate headgroups of S1P and FTY720-P substrates were modeled based on their protonation states in solution[66] since they are highly likely to interact with solvent when bound to SPNS2 in its inward-facing conformation. All substrates were then docked into the SPNS2 central cavity using Autodock Vina[67] and the pose best mimicking the DDM-bound pose were considered for simulations.

The CHARMM-GUI web-server[68] was used to set up all atomistic molecular dynamics simulations. During the setup, SPNS2 was inserted into a model membrane containing a total of 251 POPC lipids and 13 PI(4,5)$P_2$ lipids. The outer leaflet of the membrane contained only POPC lipids whereas the inner leaflet contained a 90:10 ratio of POPC:PI(4,5)$P_2$ lipids. The solvent in all simulations comprised of the TIP3P water model[69] along with 0.15 M of $Na^+$ and $Cl^-$ ions, and an additional number of $Na^+$ ions to neutralize the net charge of the entire system. All simulations used the CHARMM36m force field[70] and were performed using the GROMACS 2020.3 simulation suite[71]. Following the standard CHARMM-GUI protocol for membrane protein systems, our simulation systems were first relaxed by performing energy minimization using the steepest descent algorithm followed by several steps of equilibration during which positional restraints on the protein were gradually released. The equilibration stage was not considered for analyzes. During equilibration, the Berendsen barostat[72] was used whereas other parameter settings were maintained during production simulations and explained below. At the end of the equilibration phase for each of the three substrate-containing systems, the model membranes containing the protein and substrate measured 10 nm² along the membrane plane and 11.71 ± 0.19 nm along the perpendicular axis. These final snapshots of the equilibration phase were used to generate initial snapshots with five different initial velocities for production simulations for each substrate. Each of these five production simulations were conducted for 250 ns each with 2 fs timestep. This was performed for each substrate bound to SPNS2. The V-rescale thermostat[73] maintained the temperature at 310 K and the Parrinello-Rahman semi-isotropic barostat[74] maintained the pressure at 1 bar throughout all production simulations with a compressibility of $4.5 \times 10^{-5}$/bar. The LINCS algorithm[75] was used to apply constraints on all bond lengths. The cutoffs for Coulombic and van der Waals interaction radii were set to 1.2 nm. Van der Waals interactions were treated with a cutoff algorithm alongside a Force-switch modifier while long-range electrostatic interactions were treated by the Particle Mesh Ewald algorithm[76]. Every frame in each production simulation was generated at 100 ps intervals. Details on the simulation box dimensions and the number of lipid and solvent particles are provided in Supplementary Table 2.

The clustering analyzes of the conformations of each substrate within the SPNS2 cavity was performed with the in-built 'gmx cluster' tool of GROMACS using only the atoms of the ligand. All five simulations were concatenated and frames at regular intervals of 1 ns were used. We evaluated RMSD cutoffs for clusters of 0.2, 0.15, 0.1, 0.05, and 0.01 nm. Cutoffs larger than 0.1 nm gave narrower cluster distributions where each cluster was populated by a large number of structural configurations. The cutoffs less than 0.1 nm were unable to populate clusters. Therefore, during clustering, the Gromos method was used along with a RMSD cutoff of 0.1 nm. The coordinates of the protein-substrate complex obtained from each of the top three populous clusters were uploaded to the Protein-Ligand Interaction Profiler (PLIP) web-server[77], which then provided annotated data on different forms of protein-substrate interactions along with PyMOL-compatible

visualization states. PyMOL (https://pymol.org/2/) and Visual Molecular Dynamics (VMD)[78] were used for visualization. Xmgrace (https://plasma-gate.weizmann.ac.il/Grace/) was used for plotting.

## Reporting summary

Further information on research design is available in the Nature Portfolio Reporting Summary linked to this article.

## Data availability

The data that support this study are available from the corresponding authors upon request. The cryo-EM maps and models generated in this study have been deposited in the Electron Microscopy Data Bank (EMDB) and the Protein Data Bank (PDB), respectively. The cryo-EM maps have been deposited in the Electron Microscopy Data Bank (EMDB) under accession codes EMD-18668 (SPNS2-NbD12 in DDM), EMD-18667 (SPNS2-NbD12 in LMNG). The atomic coordinates have been deposited in the Protein Data Bank (PDB) under accession codes 8QV6 (SPNS2-NbD12 in DDM), 8QV5 (SPNS2-NbD12 in LMNG). For molecular dynamics simulations, the input files, parameters, and the initial and final structural states of all five production simulations performed for each condition have been deposited in Zenodo under accession code Zenodo-14229697 [https://doi.org/10.5281/zenodo.14229696]. The source data underlying Supplementary fig. 1b, Supplementary fig. 1c, Supplementary fig. 3b, and Supplementary fig. 9b are provided in Source Data 1. The previously published outward-facing apo SPNS2 structure was downloaded from the PDB. Source data are provided with this paper.

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

## Acknowledgements

This work was performed by the RESOLUTE (https://re-solute.eu/) and RESOLUTION (https://re-solute.eu/resolution) consortia. Plasmids are available through Addgene (https://www.addgene.org/depositor-collections/re-solute/). RESOLUTE has received funding from the Innovative Medicines Initiative 2 Joint Undertaking under grant agreement No 777372. This joint undertaking receives support from the European

Union's Horizon 2020 Research and Innovation Program and the EFPIA. REsolution has received funding from Innovative Medicines Initiative 2 Joint Undertaking under grant agreement No 101034439. This joint undertaking receives support from the European Union's Horizon 2020 Research and Innovation Program and the EFPIA. This article reflects only the authors' views and neither IMI nor the European Union and EFPIA are responsible for any use that may be made of the information contained therein. D.B.S. and A.C.W.P. were supported by the Innovative Medicines Initiative 2 Joint Undertaking (JU) under grant agreement No 875510. The JU receives support from the European Union's Horizon 2020 research and innovation program and EFPIA and Ontario Institute for Cancer Research, Royal Institution for the Advancement of Learning McGill University, Kungliga Tekniska Hoegskolan, Diamond Light Source Limited. We thank Beth MacLean for assistance with EM sample preparation and screening, Loic Carrique and Helen Duyvesteyn for EM support at OPIC, Brian Marsden for assistance with cryo-EM data processing resources, Laura Theisen and Annette Woermann for assistance with measuring activity of SPSN2 mutants, and Wyatt Yue for assistance with project management. Electron microscopy was provided through the Oxford Particle Imaging Center (OPIC), an Instruct-ERIC center (funded by Wellcome Trust JIF award [060208/Z/00/Z] and equipment grant [093305/Z/10/Z]) and the Electron Bio-Imaging Center, Diamond Light Source Ltd (eBIC; BAG proposal bi28713). The pHTBV plasmid was kindly provided by Prof. Frederick Boyce (Harvard). We thank the imaging core facility of the Medical University of Vienna for assistance with high-resolution imaging. All simulations were performed on Archer2, the national supercomputing facility via allocations provided by HECBioSim through EPSRC grant EP/X035603/1.

## Author contributions

H.Z.L., Y.N.C., C.M., A.S., G.M., S.M. cloned, expressed, and purified SPNS2. S.S. immunized the alpaca, generated the nanobody library, and identified positive binders. H.Z.L., Y.N.C., and A.S. expressed, purified, and characterized the nanobodies. Z.G. and G.W. generated knock-out, stably expressing, and transiently transfected cell lines and performed immunofluorescence experiments. H.Z.L. and A.C.W.P. collected and processed the cryo-EM images and built atomic models. H.Z.L., A.C.W.P., and D.B.S. analyzed the structures. J.S., H.C.S., F.W., H.B., and A.P.F. designed and performed the S1P transport assay. D.P. and S.K. designed and carried out the molecular dynamics simulations. A.G. and D.B.S. designed, completed, and evaluated the bioinformatics analysis. H.Z.L., A.C.W.P., and D.B.S. wrote the manuscript. All authors discussed and edited the manuscript. B.M.K., N.A.B.B., T.W., K.L.D., V.P., A.E., S.K., A.I.P., G.S.F., and D.B.S. supervised the research.

## Competing interests

J.S., F.W., H.B., and A.E. are employees of Bayer AG. Y.N.C. and V.P. are employees of Nuvisan ICB GmbH. G.S.F is co-founder and owns shares of Solgate GmbH, and SLC-focused company. The remaining authors declare no competing interests.
