## [Transparent Peer Review file · Nature Communications]

Transport and inhibition of the sphingosine-1-phosphate exporter SPNS2

Corresponding Author: Dr David Sauer

This manuscript has been previously reviewed at another journal. This document only contains information relating to versions considered at Nature Communications. Mentions of the other journal have been redacted.

Version 1:

Reviewer comments:

Reviewer #2

(Remarks to the Author)

In my initial review of this paper as submitted to [REDACTED], I posed the contrarian argument that Spns2 might be neither sufficient as an S1P transporter nor the binding site of Spns2-dependent S1P release inhibitors. The authors counter by pointing out that insofar as is known, SLC/MFS transporters act as monomers rather than part of a complex. I tend to agree with this argument.

The authors don't address the possibility that inhibitors such as 33p, which were discovered using a phenotypic assay, might not bind to Spns2. Indeed, their own data provides a test of the hypothesis that 33p binds to Spns2. For example, their modeling predicts interaction between the aromatic ring of 33p and the indole of Trp440, but the Spns2W440A mutant has the same potency for 33p (as does the Y116F mutant). The authors dismiss this negative result by invoking "plasticity" of 33p binding. Why isn't another protein binding the inhibitor an equally valid explanation?

The authors contend that 33p is competitive with S1P (lines 335-337) that, if correct, is an important finding. However, in my judgement, their data does not support that conclusion. Specifically, the variance in IC50 values (inset in Suppl Fig. 5e) is too large to support the claim of a significant change in IC50 values over the 100-fold difference in exogenous sphingosine. Further, the amount of sphingosine added to the cultures is only a surrogate for intracellular S1P. The concentration of S1P at the inner face of the plasma membrane is very difficult to know (but presumably increases with increased influx of sphingosine).

Minor Points

The author's use of the term "affinity" in reference to Spns2-dependent S1P release inhibitors is incorrect. The affinities (KI) of 16d, 33p, etc. are unknown (to my knowledge). The correct term is "potency" that, as the authors indicate, changes with formatting the biologic assay used to determine IC50 values. Specifically, it is not known whether 16d is low affinity or 33p is high affinity, rather, when compared in the same Spns2-dependent S1P release assay, 16d is considerably less potent than 33p.

I urge the authors to include the IUPAC name for 33p as a unique identifier in their methods (33p is a designation generated by J Med Chem rules for a single paper – there may be many '33p' molecules in various JMC articles).

Reviewer #3

(Remarks to the Author)

Authors sufficiently addressed my comments from the previous submission. I only have two minor concerns.

- 1) It appears the density for Glu433 is not strong enough to unambiguously model the side chain, and I would worry about showing hydrogen bonding interactions as in Supp. Fig. panel d. The density for Trp440 also do not look very convincing, but potentially sufficient to conclude the conformational change.
- 2) In line 370, DDM-bound structure is referred as substrate-bound structure. This could be misleading, and it is probably better to call the structure as DDM-bound or detergent-bound.

Reviewer Comments

Reviewer #1

No comments provided. We are grateful for their previous insights and comments, and with their apparent satisfaction with our revised manuscript.

Reviewer #2

We are very grateful for Reviewer 2's satisfaction with most of the modifications and additional work in our previously revised manuscript. The reviewer raises a few additional points that highlight holes in the field's characterization of 33p and its potential mode of action. We have addressed these in the further revised text and in the point-by-point response below.

In my initial review of this paper as submitted to [REDACTED], I posed the contrarian argument that Spns2 might be neither sufficient as an S1P transporter nor the binding site of Spns2-dependent S1P release inhibitors. The authors counter by pointing out that insofar as is known, SLC/MFS transporters act as monomers rather than part of a complex. I tend to agree with this argument.

The authors don't address the possibility that inhibitors such as 33p, which were discovered using a phenotypic assay, might not bind to Spns2. Indeed, their own data provides a test of the hypothesis that 33p binds to Spns2. For example, their modeling predicts interaction between the aromatic ring of 33p and the indole of Trp440, but the Spns2W440A mutant has the same potency for 33p (as does the Y116F mutant). The authors dismiss this negative result by invoking "plasticity" of 33p binding. Why isn't another protein binding the inhibitor an equally valid explanation?

The reviewer raises a very valid point about the poorly defined mechanism of action for 33p. We agree that our tests of 33p activity upon W440A and Y116F are unclear. However, designing mutants that disrupt only 33p activity is challenged by the apparent mobility of S1P and 33p within the central cavity, demonstrated in our MD and later experimental structures by other groups, and the significant overlap in interacting residues for both substrate and putative inhibitor. We have revised the text to reflect that our MD supports the interaction of 33p with SPNS2, but the compound's effect on the transporter's activity is under-explored.

The authors contend that 33p is competitive with S1P (lines 335-337) that, if correct, is an important finding. However, in my judgement, their data does not support that conclusion. Specifically, the variance in IC50 values (inset in Suppl Fig. 5e) is too large to support the claim of a significant change in IC50 values over the 100-fold difference in exogenous sphingosine. Further, the amount of sphingosine added to the cultures is only a surrogate for intracellular S1P. The concentration of S1P at the inner face of the plasma membrane is very difficult to know (but presumably increases with increased influx of sphingosine).

We thank the reviewer for pointing out this limitation in the significance to our examination of S1P and 33p competition. They are absolutely correct that, though a trend is present, these results are not statistically significant due to experimental limitations. We have revised the text to reflect this key point.

Minor Points

The author's use of the term "affinity" in reference to Spns2-dependent S1P release inhibitors is incorrect. The affinities (KI) of 16d, 33p, etc. are unknown (to my knowledge). The correct term is "potency" that, as the authors indicate, changes with formatting the biologic assay used to determine IC50 values. Specifically, it is not known whether 16d is low affinity or 33p is high affinity, rather, when compared in the same Spns2-dependent S1P release assay, 16d is considerably less potent than 33p.

The reviewer is quite correct in raising the essential terminological difference. We have revised the text to use potency when describing (apparent) effects on SPNS2 activity, and affinity when describing molecular interactions.

I urge the authors to include the IUPAC name for 33p as a unique identifier in their methods (33p is a designation generated by J Med Chem rules for a single paper – there may be many '33p' molecules in various JMC articles).

We are grateful to the reviewer for pointing out that including the IUPAC name would resolve any potential ambiguity in the identity of the 33p compound. This has been added to the revised methods.

Reviewer #3

We are very grateful Reviewer 3's is largely pleased with our revision in response to their previous concerns. The reviewer raises a few additional minor points that we have addressed in the newly revised text and the point-by-point response below.

Reviewer #3 (Remarks to the Author):

Authors sufficiently addressed my comments from the previous submission. I only have two minor concerns.

1) It appears the density for Glu433 is not strong enough to unambiguously model the side chain, and I would worry about showing hydrogen bonding interactions as in Supp. Fig. panel d. The density for Trp440 also do not look very convincing, but potentially sufficient to conclude the conformational change.

We appreciate the reviewer for pointing out that the density of the terminal carboxylate of Glu433 is unclear. This is a well-documented effect of the electron scattering by anionic moieties. However, the density for Glu433's C-alpha and C-beta carbons are clear in the apo structure. Similarly, the C-alpha carbon is well resolved in the DDM-bound structure, as is density for the DDM's acyl chain. Consequently, the modelled rotamer for Glu433 is the only sensible choice, as all

others clash with surrounding residues. These restrict the possible locations of the side chain's terminal carboxylate group in each structure. Furthermore, in our follow-on study (Ye et al. BioRxiv DOI: [10.1101/2024.06.13.598841](https://doi.org/10.1101/2024.06.13.598841)), we have determined a higher resolution structure of SPNS2 in DDM where the Glu433 side chain is well resolved. While this is a later, independent study whose data cannot be included here, this confirms our modeled orientation of the side chain in the DDM-bound state.

Nevertheless, we appreciate the reviewer's point that the location of the Glu433 side chain's hydrogen bonding groups cannot be precisely refined with the datasets of this study. We have revised the text and Supplementary Figure 10d to reflect this ambiguity.

2) In line 370, DDM-bound structure is referred as substrate-bound structure. This could be misleading, and it is probably better to call the structure as DDM-bound or detergent-bound.

We thank the reviewer for pointing out this misstatement. This has been corrected in the revised text.